# Optimal compression of quantum many-body time evolution operators into brickwall circuits

**Maurits S. J. Tepaske[1], Dominik Hahn[2] and David J. Luitz[1,2]**

**1** Physikalisches Institut, Universität Bonn, Nussallee 12, 53115 Bonn, Germany
**2** Max-Planck-Institut for the Physics of Complex Systems,
Nöthnitzer Straße 38, 01187 Dresden, Germany

⋆ david.luitz@uni-bonn.de

## Abstract

Near term quantum computers suffer from a degree of decoherence which is prohibitive for high fidelity simulations with deep circuits. An economical use of circuit depth is therefore paramount. For digital quantum simulation of quantum many-body systems, real time evolution is typically achieved by a Trotter decomposition of the time evolution operator into circuits consisting only of two qubit gates. To match the geometry of the physical system and the CNOT connectivity of the quantum processor, additional SWAP gates are needed. We show that optimal fidelity, beyond what is achievable by simple Trotter decompositions for a fixed gate count, can be obtained by compiling the evolution operator into optimal brickwall circuits for the $S = 1/2$ quantum Heisenberg model on chains and ladders, when mapped to one dimensional quantum processors without the need of additional SWAP gates.



## 1  Introduction

Quantum processors are a rapidly evolving technology which is expected to be pivotal for many classically hard problems like integer factorization, database search, optimization and many others [1–4]. While truly universal quantum computing is still a long shot, one of the most promising near-term applications is the simulation of complex quantum systems due to their relative similarity to the quantum hardware itself. The simulation of such systems on classical computers is extremely hard due to the exponential complexity in terms of storage and computer time, while both problems are naturally solved on quantum hardware.

There are two different approaches: analog and digital quantum simulators. Analog simulators are specifically engineered systems to mimic the corresponding dynamics of the target system and are often based on quantum optical setups. This technique has been successfully applied to condensed matter systems [4–9] and lattice gauge theories [10–12] and is in principle extremely powerful but requires a tailored experimental setup for a given type of problem.

In contrast, digital quantum simulators [13] rely on a discrete representation of the wave function on an array of two level systems (dubbed qubits), which can be fully controlled by a universal set of quantum gates which allows in principle for the representation of any unitary operation on the many-body wave function, represented as a sequence of gates. Due to the universal representation of the wave function, this is an attractive approach which is extremely flexible once a suitable mapping of the system of interest to qubits is devised. Recent applications include condensed matter systems [14–19], simulations from quantum chemistry [11,20–22] and high-energy physics [23,24]. Digital quantum simulations were also used to realize exotic phases of matter like time crystals [25,26] and quantum spin liquids [27].

The state-of-the-art method for simulating the real time dynamics of complex quantum systems involves a factorization of the time evolution operator into a sequence of gates using Trotter decompositions of different orders [28–32], introducing discrete time steps to get an approximation of the exact time evolution of the system. This introduces a discretization error, which can be systematically controlled by using smaller step sizes. As a downside, small step sizes require a larger number of gates. Due to the fragility of the quantum state stored in the machine, and due to hardware imperfections, each additional gate potentially introduces new sources of error due to dissipation processes. Hence a trade-off between discretization errors and errors due to intrinsic machine noise during the simulation is required. To achieve optimal fidelity in light of this tradeoff, it is therefore important to minimize the resource costs for a given simulation. Recent work yielded tighter bounds for the discretization errors [33]. Furthermore, it was also argued recently that beyond a certain step size the fidelity of the Trotter decomposition breaks down in a universal fashion, leading to a regime of quantum chaos [34,35]. This sets also upper bounds for possible step sizes. It remains however unclear,

whether better alternatives to Trotter decompositions exist.

One promising approach in this regard are quantum variational algorithms. The main idea of them is to approximate a time-evolved state using a parametrized circuit [36–39]. The parameters are then fixed using optimization algorithms on a quantum computer. Recent numerics suggest that the number of parameters needed to describe time-evolved states or ground states scales favorable even in comparison to matrix-product states [40, 41]. Most of these algorithms involve optimization where gradients are measured directly on the quantum devices, or they use deep learning approaches. However, the measurement of gradients on a quantum device is currently infeasible due to the high error rates, while optimization using deep neural networks is not controlled.

In this paper we take a more universal approach. Rather than focussing on the wave function, we directly target the time evolution operator, aiming at a compact representation as a shallow circuit. We use brickwall circuits in which the gates are parametrized two qubit unitaries, connecting neighboring qubits in the architecture of the quantum processor as an ansatz for the time evolution operator. This parametrized circuit can be optimized classically to represent the time evolution operator for a given time step with high fidelity. The resulting circuit can then be repeated to evolve the quantum state to later times. We show that such an optimized circuit can yield significantly higher fidelity time evolution for a fixed gate count compared to the traditional Trotter decomposition and is thus superior for digital quantum simulation.

We also show that this strategy allows us to obtain similar accuracy using significantly less gates, even for systems where the physical geometry does not coincide with the proposed circuit architecture, essentially "baking in" the otherwise required SWAP gates to match geometries into the circuit. As an interesting benchmark problem, we use our approach to compute out-of-time-ordered correlators (OTOCs) and show that we achieve better accuracy than Trotter methods with similar resource cost. Finally, we analyze the gate structure of the optimized gates, as a first step towards further improvements of this approach.

## 2 Model and Method

### 2.1 Model

For concreteness and simplicity, we focus on simulating finite systems of $s = 1/2$ spins on a lattice with $L$ sites, designed to be performed on a quantum processor with an identical Hilbert space $\mathcal{H}$, which is the product space of $L$ two-level quantum systems (qubits) $\bigotimes_{i=1}^{L} \mathcal{Q}_i$ and has an exponentially growing dimension $\dim H = 2^L$. Specifically, we discuss spin-1/2 systems with SU(2) symmetric Heisenberg couplings

$$h_{ij} = \sigma_i^x \sigma_j^x + \sigma_i^y \sigma_j^y + \sigma_i^z \sigma_j^z, \tag{1}$$

between nearest neighbor (NN) spins on a chain $\mathfrak{c}$ and a triangular ladder $\mathfrak{l}$, both with open boundary conditions, i.e.

$$H_{\mathfrak{c}} = \sum_{\langle i,j \rangle} h_{ij}, \qquad H_{\mathfrak{l}} = \sum_{\langle\langle i,j \rangle\rangle} h_{ij}. \tag{2}$$

Here $\sigma_{x,y,z}$ are the usual Pauli matrices while $\langle i, j \rangle$ and $\langle\langle i, j \rangle\rangle$ denote the NN sites of the chain, or the NN sites of our triangular ladder geometry (note that this is identical to a chain with nearest and next nearest neighbor (NNN) interactions). These lattice geometries are illustrated in Fig. 1.

Most current quantum devices using superconducting qubits are not capable of all-to-all connectivity, i.e. due to the chip setup two qubit gates can only be applied between neighboring

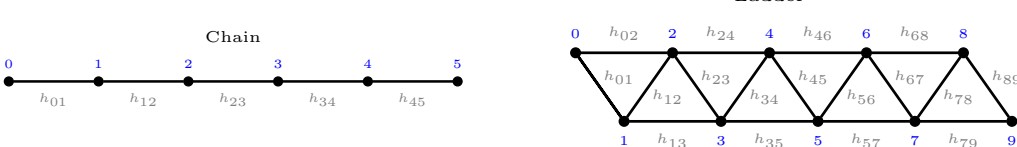

Figure 1: The chain (left) and triangular ladder (right) lattice geometries used in this work.

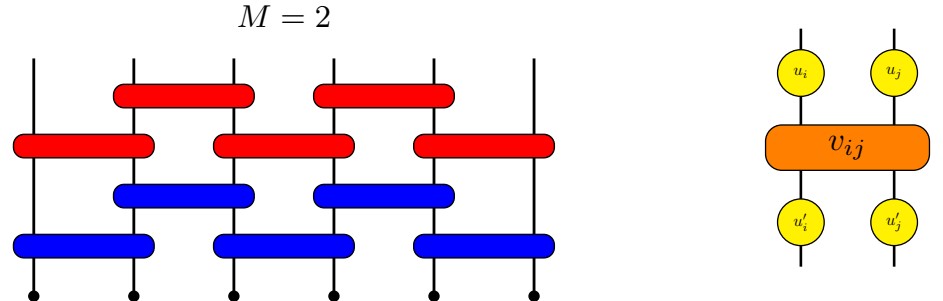

Figure 2: Left: A brickwall circuit with depth $M = 2$ for six qubits, with each color representing a $M = 1$ layer. Circles represent the initial state of the qubits and boxes indicate a two qubit unitary gate applied to a pair of neighboring qubits. Right: Parametrization of a two qubit unitary as a product of four single qubit gates and one two qubit gate.

qubits, which are arranged in different geometries [42–44] In order to apply gates between distant qubits, one has to use a sequence of swap gates, which exchange the quantum state of neighboring qubits, such that effectively the states of distant qubits are moved to neighboring qubits in the processor geometry. On these, any two qubit gate can be applied and then the swap sequence needs to be applied in reverse order. This requires a great number of additional gates and therefore introduces further possible sources of errors.

Our goal is therefore to find the best unitary circuit $\mathcal{C}$ of a given depth $M$ to approximate the time evolution operator $\mathcal{U}(t) = \exp(-itH_{c/l})$. In order to mimic the limited connectivity of current quantum devices, we choose $\mathcal{C}$ to consist only of NN two-qubit gates on a 1d chain, arranged in a brickwall pattern, i.e. we model our quantum processor as an open chain of qubits, while one of our physical models we want to simulate on this machine has a different, triangular ladder, geometry. This allows us to investigate whether it is possible to compile the time evolution operator in a nearest neighbor, brickwall circuit (exemplified in the left panel of Fig. 2) without the need for additional swap gates, which are generally costly on superconducting platforms.

## 2.2 Trotter circuits

To benchmark the performance of the brickwall circuits we will compare them with the first-, second- and fourth-order Trotter circuits that are based on the well known Trotter decompositions [45]. Here we introduce these circuits for the Hamiltonians (2) that are used in this work.

For the chain Hamiltonian $H_c$ we have two non-commuting parts, namely the bond Hamil-

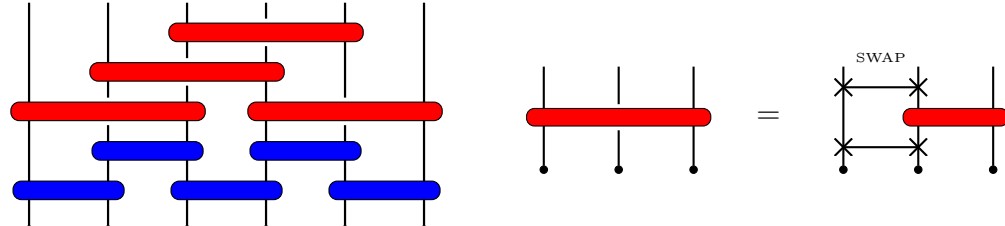

Figure 3: Left: The blue brickwall layer encodes the first-order Trotter decomposition for NN interacting Hamiltonians. The combination of the blue and red layers encodes the first-order Trotter decomposition for NNN interacting Hamiltonians, where the blue gates act on NN qubits whereas the red gates act on NNN qubits. Right: The decomposition involving SWAP gates, displayed as the crossed line, which is used to convert the NNN two-qubit gate into a circuit involving only two-qubit gates.

tonians $h_{i,i+1}$ (1) on alternating bonds, such that we can split $H_{\mathfrak{c}}$ in two commuting parts as

$$H_{\mathfrak{c}} = H_1 + H_2 = \sum_{i=0,2,\dots} h_{i,i+1} + \sum_{i=1,2,\dots} h_{i,i+1}\,. \tag{3}$$

For the ladder Hamiltonian we have on top of this three extra non-commuting parts due to the NNN couplings, i.e. we can split $H_{\mathfrak{l}}$ into five commuting parts as

$$H_{\mathfrak{l}} = H_{\mathfrak{c}} + H_3 + H_4 + H_5 = H_{\mathfrak{c}} + \sum_{i=0,3,\dots} h_{i,i+2} + \sum_{i=1,4,\dots} h_{i,i+2} + \sum_{i=2,5,\dots} h_{i,i+2}\,. \tag{4}$$

By writing the Hamiltonians in this way we can define the $M = 1$ first-order Trotter circuits for $H_{\mathfrak{c}}$ and $H_{\mathfrak{l}}$ as [45]

$$\mathcal{U}_{\mathfrak{c}}^{\text{1st}}(t) = \mathcal{U}_1(t)\mathcal{U}_2(t) \tag{5}$$

$$= \exp(-\mathrm{i}tH_1)\exp(-\mathrm{i}tH_2)\,, \tag{6}$$

$$\mathcal{U}_{\mathfrak{l}}^{\text{1st}}(t) = \mathcal{U}_1(t)\mathcal{U}_2(t)\mathcal{U}_3(t)\mathcal{U}_4(t)\mathcal{U}_5(t) \tag{7}$$

$$= \exp(-\mathrm{i}tH_1)\exp(-\mathrm{i}tH_2)\exp(-\mathrm{i}tH_3)\exp(-\mathrm{i}tH_4)\exp(-\mathrm{i}tH_5)\,. \tag{8}$$

These circuits approximate the exact $\mathcal{U}(t) = \exp(-\mathrm{i}tH_{\mathfrak{c}/\mathfrak{l}})$ with error $\mathcal{O}(t^2)$ [33]. Note that depth $M = 1$ for the Trotter circuits does not mean one brickwall layer, but instead one Trotter step $\mathcal{U}_{\mathfrak{c}/\mathfrak{l}}^{\text{1nd}}(t)$. While these coincide for the first-order Trotter circuit for the chain, this is not the case for the first-order Trotter circuit for the ladder, and for the second- and fourth-order Trotter circuits which we introduce below. The circuit diagram for $\mathcal{U}_{\mathfrak{c}}^{\text{1st}}(t)$ is shown as the blue brickwall layer in the left panel of Fig. 3, where $\mathcal{U}_1(t)$ is the half-brickwall layer on odd bonds and $\mathcal{U}_2(t)$ is the half-brickwall layer on even bonds. The circuit diagram for $\mathcal{U}_{\mathfrak{l}}^{\text{1st}}(t)$ is the full circuit in this figure, where $\mathcal{U}_1(t)$ and $\mathcal{U}_2$ again form the blue brickwall layer while $\mathcal{U}_3(t), \mathcal{U}_4(t)$ and $\mathcal{U}_5(t)$ form the red layer, containing two-qubit gates that act on NNN instead of NN qubits. To turn this into a circuit that involves only NN two-qubit gates we introduce the SWAP gate and decompose every NNN gate as in the right panel of Fig. 3.

The circuit layers $\mathcal{U}_1, \mathcal{U}_2, \mathcal{U}_3, \mathcal{U}_4, \mathcal{U}_5$ form the building blocks of the second- and fourth-order Trotter circuits. The $M = 1$ second-order Trotter circuits are composed as [45]

$$\mathcal{U}_{\mathfrak{c}}^{\text{2nd}}(t) = \mathcal{U}_1(t/2)\mathcal{U}_2(t)\mathcal{U}_1(t/2)\,, \tag{9}$$

$$\mathcal{U}_{\mathfrak{l}}^{\text{2nd}}(t) = \mathcal{U}_1(t/2)\mathcal{U}_2(t/2)\mathcal{U}_3(t/2)\mathcal{U}_4(t/2)\mathcal{U}_5(t)\mathcal{U}_4(t/2)\mathcal{U}_3(t/2)\mathcal{U}_2(t/2)\mathcal{U}_1(t/2)\,, \tag{10}$$

which approximate the exact evolution operators with error $\mathcal{O}(t^3)$ [33]. Using these second-order Trotter circuits we can define the $M = 1$ fourth-order Trotter circuits as [45]

$$\mathcal{U}_{\mathfrak{c}/\mathfrak{l}}^{\text{4th}}(t) = \mathcal{U}_{\mathfrak{c}/\mathfrak{l}}^{\text{2nd}}(t_1)\mathcal{U}_{\mathfrak{c}/\mathfrak{l}}^{\text{2nd}}(t_1)\mathcal{U}_{\mathfrak{c}/\mathfrak{l}}^{\text{2nd}}(t_2)\mathcal{U}_{\mathfrak{c}/\mathfrak{l}}^{\text{2nd}}(t_1)\mathcal{U}_{\mathfrak{c}/\mathfrak{l}}^{\text{2nd}}(t_1), \tag{11}$$

where we defined the time steps

$$t_1 = \frac{1}{4 - 4^{1/3}}t, \qquad t_2 = (1 - 4t_1)t. \tag{12}$$

These circuits approximate the exact evolution operators with error $\mathcal{O}(t^5)$ [33].

Because we are concerned with circuits that are implemented on a quantum processor with only NN qubit connectivity, we have to convert every NNN two-qubit gate that appears in $\mathcal{U}_{\mathfrak{l}}^{\text{1st}}, \mathcal{U}_{\mathfrak{l}}^{\text{2nd}}, \mathcal{U}_{\mathfrak{l}}^{\text{4th}}$ to three NN two-qubit gates, as shown in the right panel of Fig. 3. The gate counts $N_g$ of the resulting NN Trotter circuits are given in Sec. A, also for the chain geometry.

## 2.3 Optimization

Each two-qubit gate $U_{ij} \in \mathbb{C}^{4\times4}$ of the circuit $\mathcal{C}$, acting on two neighboring qubits $i$ and $j$, can be decomposed into a product of one-qubit gates $u_i \in \mathbb{C}^{2\times2}$ and a two-qubit gate $v_{ij} \in \mathbb{C}^{4\times4}$ [46]

$$U_{ij} = (u_i \otimes u_j)v_{ij}(u_i' \otimes u_j'). \tag{13}$$

Here $v_{ij}$ is parameterized as

$$v_{ij}(\lambda_0, \lambda_1, \lambda_2) = e^{-i(\lambda_0\sigma_i^x\otimes\sigma_j^x + \lambda_1\sigma_i^y\otimes\sigma_j^y + \lambda_2\sigma_i^z\otimes\sigma_j^z)}, \tag{14}$$

with three real parameters $\lambda_{0,1,2}$, and the $u_i$ are parameterized up to a global phase as

$$u_i(\phi_0, \phi_1, \phi_2) = \begin{pmatrix} e^{i\phi_1}\cos(\phi_0) & e^{i\phi_2}\sin(\phi_0) \\ -e^{-i\phi_2}\sin(\phi_0) & e^{-i\phi_1}\cos(\phi_0) \end{pmatrix}, \tag{15}$$

each containing three real parameters $\phi_{0,1,2}$. Hence this decomposition of $U_{ij}$ contains 15 real parameters, and it can be visualised as in the right panel of Fig. 2. To represent the unitary gate as a global unitary matrix, acting on the full wave function, we introduce its matrix form

$$\text{mat}(U_{ij}) = I_{2^{i-1}} \times U_{ij} \times I_{2^{L-j}}, \tag{16}$$

by taking the Kronecker product with identity matrices on the qubits on which the gate does not act (and implicitly encoding the nearest neighbor condition $j = i + 1$). The entire circuit is a product of such unitaries and can formally be expressed by

$$\mathcal{C} = \prod_{k=0}^{N_g} \text{mat}(U_{i_k,j_k}), \tag{17}$$

where $N_g$ is the total number of gates in the circuit. Since each gate is parametrized by $\vec{\theta}_{i_k} = (\vec{\lambda}_{i_k}, \vec{\phi}_{i_k})$, the circuit depends on all these $15N_g$ parameters $\vec{\theta} = (\vec{\theta}_{i_0}, \vec{\theta}_{i_1}\dots) \in \mathbb{R}^{15N_g}$

$$\mathcal{C}(\vec{\theta}) = \prod_{k=0}^{N_g} \text{mat}(U_{i_k,j_k}(\vec{\theta}_{i_k})). \tag{18}$$

In practice, when stacking the gates to form the circuit, we merge two one-qubit unitaries into a single one-qubit unitary where possible, since a product of general one-qubit unitaries can be written as a single general one-qubit unitary. This reduces the amount of circuit parameters.

We would like to find an optimal parameter set $\vec{\theta}$ for a given circuit architecture, such that the distance between the unitary represented by the circuit $\mathcal{C}(\vec{\theta})$ and the targeted time evolution operator $\mathcal{U}$ of the system up to time $t$ is minimized. For two unitary operators $\mathcal{U}$ and $\mathcal{C}$, we therefore define a measure of distance in terms of the normalized Frobenius norm, namely the "infidelity" $\epsilon$, given by

$$\epsilon = \frac{1}{2}\frac{\|\mathcal{U}-\mathcal{C}\|_F^2}{2^L} = \frac{1}{2^{L+1}}\mathrm{Tr}\big[(\mathcal{U}-\mathcal{C})^\dagger(\mathcal{U}-\mathcal{C})\big] = 1 - \frac{\mathrm{ReTr}\big[\mathcal{U}^\dagger\mathcal{C}\big]}{2^L}. \tag{19}$$

We use this infidelity as an objective function, such that we obtain a minimization problem for a fixed circuit architecture (number and sequence of two qubit gates). In our case the target unitary $\mathcal{U}$ is an approximation of an exact time-evolution operator, where the error stems from the tensor network methods that make the optimization tractable.

The objective function $\epsilon$ needs to be evaluated many times during the optimization and we find that it is efficient to first compress the time evolution operator $\mathcal{U}$ into a matrix product operator (MPO) $\mathcal{M}_\chi$ of bond dimension $\chi$, such that we can calculate $\epsilon$ via efficient standard tensor network methods. For the local systems we investigate here and for short times, this is always efficient, due to the low operator entanglement of the time evolution operator [47]. In particular, we discard the smallest singular values for which the squares sum to a tiny number, since their contribution is negligible, such that lowly entangled operators do not saturate the maximum bond dimension $\chi$. To obtain the (truncated) MPO representation of $\mathcal{U}$ with negligible discretization error, we take an identity MPO and perform time-evolving block decimation [45, 48] with a small timestep $\delta t = 10^{-4}$ and fourth-order Trotter decomposition, such that the introduced error is negligible.[1]

To optimize the parameters $\vec{\theta}$ of the circuit such that $\epsilon$ is minimal, we employ the paradigm of differentiable programming [49]. Here the gradient $\nabla_{\vec{\theta}}\,\epsilon$ is calculated in a similar fashion as the original backpropagation algorithm used for deep neural networks [50], which has been generalized to arbitrary programs, including tensor network algorithms [49]. To this end, a program is represented as a computational graph through which the local gradients are propagated, which requires each computational component to have a well-defined gradient. In particular, for the tensor network algorithm in this work, the SVD is a crucial component, and so it is important to construct a stable SVD gradient [49]. Fortunately, differentiable programming inherits the cost from its base algorithm, i.e. in our case from the $M$ SVDs that are performed when obtaining the circuit MPO at every iteration. As a result our algorithm has the scaling $\mathcal{O}(N_i L M d^6 \chi^3)$, where $N$ is the amount of gradient descent iterations. Importantly, even though the cost scales linearly with system size $L$ and circuit depth $M$, the amount of parameters grows as $\mathcal{O}(LM)$, such that the amount of iterations required to reach a low-lying minimum also grows, because local minima prolifrate with growing parameter count [51].

Using the global gradient $\nabla_{\vec{\theta}}\,\epsilon$ we then perform gradient descent. We use this global optimization procedure instead of the local optimization from [40] because we found that this yields significantly higher fidelity when an Adam-like adaptive learning rate is used [52]. Here it is crucial not to stop optimizing when the infidelity appears to have stagnated, since we have often found that the optimization gets stuck in such a "local minimum" for some time before it jumps out and converges to a lower minimum. This is possibly related to the "barren plateau" problem that often occurs when performing gradient descent for quantum circuits with a large parameter space, where the optimization reaches a set of circuit parameters for which the majority of its gradients become very small such that the optimization (temporarily) halts [53]. In Sec. B we review the Adam method and discuss the mentioned convergence behavior in more detail.

---

[1] We compare the results for our circuits to Trotter circuits with comparable gate counts, and in all instances of the involved Trotter circuits, the timesteps are several orders of magnitude larger than the stepsize used to approximate the target unitary $\mathcal{U}$.

At small $M$ the optimized circuits in a sense compress the targeted time evolution operator, especially when its time-step is large, and therefore they are called "compressed circuits". In Sec. D we check if the lattice symmetries of the targeted unitary emerge in the gates of the optimized circuits.

## 2.4 Stacking circuits

The general strategy we implement is the following: For some (short) timestep $t$, we find an optimal circuit $\mathcal{C}(\vec{\theta})$ which best approximates the targeted time evolution operator $U_t$. In principle, $t$ is arbitrary, with the general logic that shorter $t$ unitaries can be encoded by shallower circuits (lower $M$). In practice, $t$ will be also governed by the time grid, on which observables should be evaluated, although this could be achieved also by working with two or more different optimized circuits with different $t$, a case we do not further discuss in this work. To propagate the wave function to longer times, which are multiples of $t$, we then use the circuit

$$\mathcal{C}(\vec{\theta})^n \approx \mathcal{U}_t^n. \tag{20}$$

It is interesting to investigate how well this stacked circuit performs for time evolution to longer times and we will confront these results to benchmarks for the circuits discussed in Sec. 2.2 that result from traditional Trotter decompositions.

## 2.5 Quantities of interest

Having obtained the compressed circuits for short times, for which the relatively low entanglement allows for an accurate description with truncated MPOs, we then compute $\epsilon$ for long times using the stacked circuits as approximation. If we now were to use the same MPO formalism that was used during the optimization, the growing of entanglement as we stack the circuit multiple times results in either an unfeasible amount of required computational resources or significant truncation errors. In particular, the stacked circuit represents a target unitary at large times, which generally has large entanglement, such that an accurate MPO representation requires a saturated bond dimension, i.e. the central tensors would require bond dimension $2^L$ to prevent significant truncation errors.

For a highly entangled MPS $|\psi_i\rangle$ this central bond dimension is instead $2^{L/2}$, which is still managable for the system sizes considered in this work. Hence, to probe the true representablity of the stacked circuit, without having to deal with artefacts of the tensor network method, we use typicality [54]. Here the trace in Eq. 19 is replaced by the average over $N_\psi$ Haar random states $|\psi_i\rangle$, i.e.

$$\text{Tr}\left[\mathcal{U}^\dagger \mathcal{C}\right] \approx \frac{1}{N_\psi} \sum_i \langle \psi_i | \mathcal{U}^\dagger \mathcal{C} | \psi_i \rangle. \tag{21}$$

This allows us to calculate $\epsilon$ in an unbiased manner for the system sizes considered in this work.

Besides using the infidelity $\epsilon$ as a measure of the performance of the circuits, we will also use the circuits to compute out-of-time-ordered correlators (OTOCs) [55]. For spin-1/2 $\sigma^z$ operators, the OTOC $C_{ij}$ between lattice sites $i$ and $j$ is defined with the Frobenius norm as

$$C_{ij}(t) = \left\| \left[ \sigma_i^z(t), \sigma_j^z \right] \right\|_F^2, \tag{22}$$

where $\sigma_i^z(t) = \mathcal{C}^\dagger \sigma_i^z \mathcal{C}$ is the spin operator on site $i$ evolved by the circuit. As for the infidelity, it is important to use typicality instead of the truncated MPO formalism when calculating $C_{ij}$ for a circuit that is stacked many times.

To calculate (22) we invoke the hermiticity of the spin operators $\sigma^z$, such that by expanding the commutator in (22) we can write the OTOC as

$$C_{ij}(t) = 1 - \frac{1}{4}\text{Tr}\big[\sigma_j^z \sigma_i^z(t) \sigma_j^z \sigma_i^z(t)\big], \tag{23}$$

which is readily calculated in the MPO formalism. Concretely, we take an identity MPO and put a $z$-spin operator $\sigma^z$ at site $i$, which is then evolved in the Heisenberg picture by the circuit $\mathcal{C}$, yielding a different MPO. Then we again take an identity MPO and put a $z$-spin operator on site $j$, which we do not evolve. Then we calculate the trace in (23) via a full contraction of four MPOs, which can be done efficiently.

## 3 Results

To benchmark the performance of the compression strategy outlined in Sec. 2, we systematically analyze the infidelity $\epsilon$ as a function of simulation time step $t$, total gate count $N_g$ and system size, in direct comparison to Trotter decompositions of different orders, and present these results in Sec. 3.1. In Sec. 3.2 we extend this systematic analysis to out-of-time-ordered correlators (OTOCs) (22). Furthermore, in Sec. 3.3 we probe the structure of the gates that make up the optimized circuits, in an attempt to uncover the structures that allow these circuits to outperform their Trotter counterparts.

### 3.1 Infidelity

As a first test of the circuit optimization algorithm outlined in Sec. 2, we compare the optimal infidelities of compressed circuits to those of comparable Trotter circuits. Concretely, we consider time evolution operators of the chain and ladder Heisenberg Hamiltonians (2) at three system sizes $L = 8, 12, 16$ and two time-steps $t = 1, 2$. For each Hamiltonian, system size and time-step, we determine the time evolution operator $\mathcal{U}$ with numerically negligible discretization error for a certain bond dimension $\chi$, and perform the global optimization as outlined in Sec 2 to minimize the infidelity $\epsilon$ of the compressed circuit. For $L = 8, 12, 16$ we have taken $\chi = 256, 150, 100$ as a compromise between precision and practical efficiency. We note that our main concern here is not to get a numerically exact MPO representation, but rather a reasonably good approximation of the time evolution operator. We call this our target time-evolution operator, which we want to approximate with our circuits.

As a first benchmark, we take for each of our parameter sets various circuit depths $M = 1, 2, 4, 8, 16$, where $M$ is the number of elementary layers of $L - 1$ gates, and consider $\epsilon$ as a function of the corresponding gate count $N_g$ (see Sec. A for details on how to obtain the number of gates). We compare this with first-, second- and fourth-order Trotter circuits [45].

The results are shown in Fig. 4. The left pair of panel columns is for the chain and the right pair is for the ladder. The first and third columns are for time-step $t = 1$ and the second and fourth are for $t = 2$. The upper row is for system size $L = 8$, the middle row is for $L = 12$, and the bottom row is for $L = 16$. Each panel contains the infidelities of the optimized compressed circuits (CC) as a red line, and the infidelities of the Trotter circuits as blue lines. The infidelities of the Trotter circuits are calculated for the same depths $M$ as the compressed circuit, where it should be remembered from Sec. 2.2 that in this case $M$ is not necessarily equal to the amount of brickwall layers in the Trotter circuit, but is instead equal to the amount of Trotter steps that compose the circuit. The time-step of the Trotter step is chosen as $t/M$, such that $M$ subsequent steps correspond to a total time-step $t$. The gate counts of the Trotter circuits were calculated with the expressions in Sec. A, which take into account the number of swap gates required to map the ladder geometry to a chain of qubits.

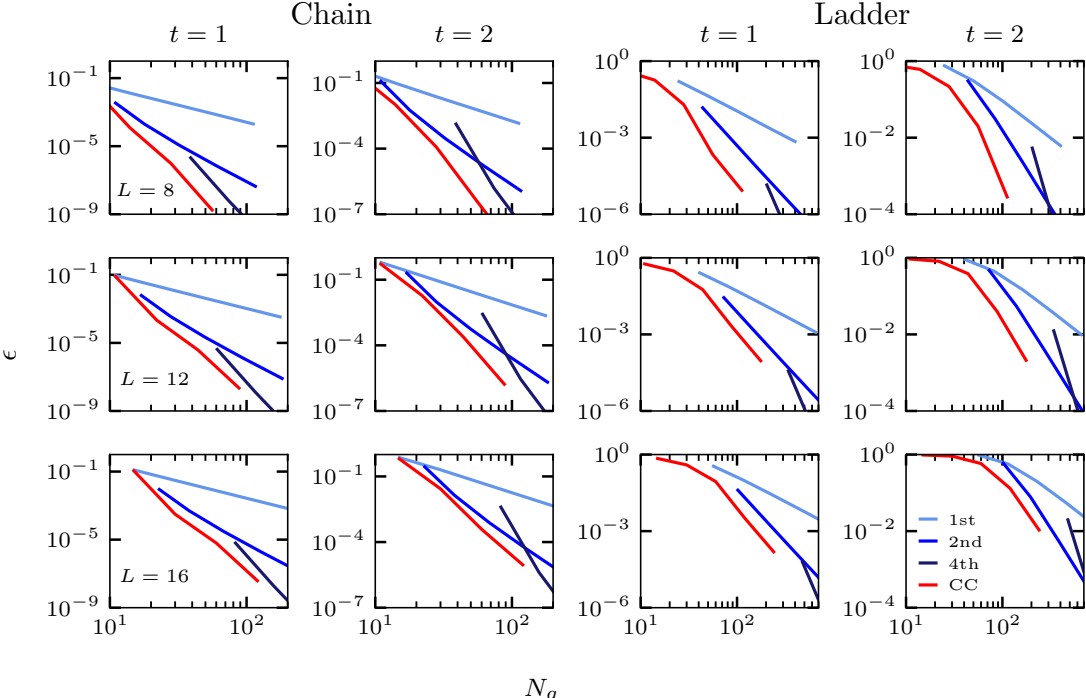

Figure 4: The infidelity $\epsilon$ versus gate count $N_g$ for the time evolution operator of the Heisenberg model on a chain (left panels) and ladder (right panels) in log-log scale. The first and third columns are for $t = 1$ while the second and fourth columns are for $t = 2$. The top panels are for $L = 8$ and a time evolution MPO with $\chi = 256$, the middle panels are for $L = 12$ with $\chi = 150$, and the bottom panels are for $L = 16$ with $\chi = 100$. The blue curves represent the Trotter circuits and the red curve represents the compressed circuit (CC).

From Fig. 4 it becomes clear that per gate the compressed circuit outperforms the Trotter circuits for all considered parameter sets. Moreover, it appears that for $L = 8$ the infidelity of the compressed circuit roughly scales with $N_g$ like the best Trotter order, but with a more favorable prefactor, i.e. at intermediate gate counts it scales as second-order whereas at the highest probed gate count it scales as fourth-order. We have found that the same picture emerges when plotting $\epsilon$ versus the $t$ at which the circuit was optimized, where $M = 1$ scales like first-order Trotter, and by increasing $M$ we approach the fourth-order scaling, passing through the second-order scaling. We expect the same to hold for $L = 12$ and $L = 16$, if we could reach a lower minimum, but here the optimization is more expensive.

Having considered the infidelities of the compressed circuits at the time-step for which they were optimized, we now quantify how these infidelities grow when the circuits are stacked, which we do for the same systems as in Fig. 4. To this end we select a compressed circuit that was optimized at $t = 2$, and take for every Trotter order a circuit of depth $M$ with a gate count as close as possible to that of the compressed circuit, and choose its time-step to be $t/M$.

Concretely, for the chain we take a compressed circuit with $M = 8$, in which case we have to take first-, second-, and fourth-order Trotter circuits with $M = 8, 7, 1$. Using the gate count equations from Sec. A we find that for $L = 8$ the circuits have $N_g = 56, 56, 53, 39$, for $L = 12$ they have $N_g = 88, 88, 83, 61$, and for $L = 16$ they have $N_g = 120, 120, 113, 83$. For the ladder we take a compressed circuit with $M = 16$, such that we have to take first-, second-, and fourth-order Trotter circuits with $M = 4, 3, 1$. The corresponding gate counts are $N_g = 112, 100, 124, 204$ for $L = 8$, $N_g = 176, 164, 207, 341$ for $L = 12$,

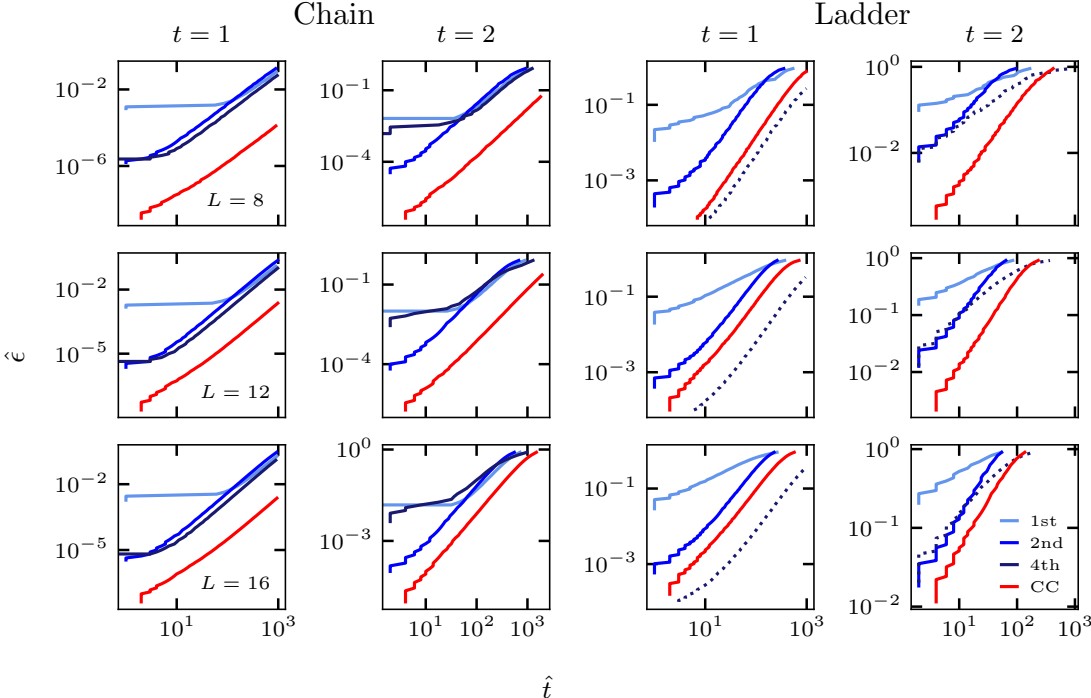

Figure 5: The time $\hat{t}$ after which the stacked circuits exceed the infidelity threshold $\hat{\epsilon}$, for the time evolution operator of the Heisenberg model on a chain (left panels) and ladder (right panels) in log-log scale. The first and third columns are for circuits optimized at $t = 1$ while the second and fourth columns are for $t = 2$, with the circuits being stacked up to a thousand times. The circuits were chosen such that they have similar gate counts, with $M = 8, 8, 7, 1$ for the chain and $M = 16, 4, 3, 1$ for the ladder, for the compressed circuit and first-, second- and fourth-order Trotter circuits, respectively. The top panels are for $L = 8$ with $\chi = 256$, the middle panels are for $L = 12$ with $\chi = 150$, and the bottom panels are for $L = 16$ with $\chi = 100$. The blue curves represent the Trotter circuits and the red curve represents the compressed circuit (CC). The fourth-order Trotter circuit for the ladder is displayed as a dashed line, since it contains roughly twice as many gates as the compressed circuit and is therefore not necessarily indicative of their relative performance.

and $N_g = 240, 228, 290, 478$ for $L = 16$.

  To quantify the quality of the compressed and Trotter circuits under stacking, we take various infidelity thresholds $\hat{\epsilon}$ and stack the circuits up to a thousand times until they cross this threshold at some time $\hat{t}$, i.e. we determine $\epsilon(\hat{t}) = \hat{\epsilon}$. As mentioned in Sec. 2 we utilize typicality (21) to calculate the stacked infidelities.

  In Fig. 5 we plot $\hat{\epsilon}$ versus $\hat{t}$ in log-log scale. The used color coding is identical to that of Fig. 4, except that the fourth-order Trotter circuit for the ladder is now represented with a dashed line, to emphasize that its infidelity relative to that of the compressed circuit is not necessarily indicative of the relative performance, because it contains roughly twice as many gates as the compressed circuit. From these plots it is clear that the advantage of the compressed circuits from Fig. 4 is not lost when stacking it many times. In particular, in all considered cases the compressed circuits are able to go to significantly larger times, at all infidelity thresholds, than the Trotter counterparts. The only exception is for the ladder at $t = 1$, where the fourth-order Trotter circuit performs better, but as mentioned this Trotter circuit has twice as many gates as the compressed circuit and is therefore not a fair comparison.

From the plots we extract the universal quadratic power-law $\hat{\epsilon} \propto \hat{t}^2$, for both the compressed and the Trotter circuits. This error scaling is analogous to first-order Trotter decomposition. The only exception is the ladder with $L = 16$ at $t = 2$, where the infidelity reaches $\epsilon \approx 1$ rather quickly, such that it is situated in the rounding part that is also observed for the $t = 1$ ladder curves at the high-infidelity end. The gap between the compressed circuits and the best performing Trotter circuits is thus found to grow quadratically with $\hat{t}$. Concretely, for the chain with $L = 12$ and timestep $t = 1$, we find that for $\hat{\epsilon} = 10^{-3}$ the compressed circuit has $\hat{t} = 644$ whereas the best Trotter circuit (i.e. of fourth-order) has $\hat{t} = 94$. For $\hat{\epsilon} = 10^{-4}$ we instead get $\hat{t} = 201$ for the compressed circuit and $\hat{t} = 29$ for the best Trotter circuit. For the same system at timestep $t = 2$, we find that at $\hat{\epsilon} = 10^{-3}$ the compressed circuit has $\hat{t} = 116$ while the best Trotter circuit has $\hat{t} = 14$. At $\epsilon = 10^{-2}$ we have $\hat{t} = 378$ for the compressed circuit and $\hat{t} = 46$ for the best Trotter circuit. From these values it is clear that for the chain we can go roughly eight times further in time than the best Trotter circuit with similar gate count. These values are for $L = 12$, and the same analysis at $L = 8$ reveals that here we can go fourteen to twenty times as far, while for $L = 16$ we can go three to eight times as far, with the lower bounds for $t = 2$ and the upper bounds for $t = 1$. These values emphasize that the larger we choose $\hat{\epsilon}$, the larger the gap between $\hat{t}$ of the compressed and Trotter circuits becomes, which grows quadratically as stated above. This implies that the superiority of the compressed circuits over Trotter circuits becomes especially apparent when we set a relatively high error threshold, which for the compressed circuits is reached at much larger time than for Trotter circuits which have comparable gate count.

Repeating this analysis for the ladder, again starting off with $L = 12$ and $t = 1$, we find at $\hat{\epsilon} = 10^{-2}$ that the compressed circuit has $\hat{t} = 34$ whereas the best Trotter circuit, excluding the fourth-order Trotter with double the gate count, has $\hat{t} = 14$. With $\hat{\epsilon} = 10^{-1}$ the compressed circuit has $\hat{t} = 125$ whereas the second-order Trotter circuit has $\hat{t} = 57$. For the same system at $t = 2$ and with $\hat{\epsilon} = 10^{-1}$, we have $\hat{t} = 40$ for the compressed circuit and $\hat{t} = 10$ for the second-order Trotter circuit. Hence for the ladder we can go roughly two to four times as far than the best Trotter circuit with comparable gate count. Repeating this analysis for $L = 8$ we find that we can go five to two times farther, and for $L = 16$ we can go three to two times farther, again with the lower bounds for $t = 1$ and the upper bounds for $t = 2$.

Instead of examining the stacking behavior of compressed and Trotter circuits with comparable gate count, we now compare how circuits with comparable optimized infidelity stack, to see whether similar fidelities are achievable with compressed circuits that have only a fraction of the gates of Trotter circuits. To this end we consider the chain and ladder for a single system size $L = 12$, with time-step $t = 2$ for the chain and $t = 1$ for the ladder, and we stack the circuits up to $t = 20$. For simplicity we compare only with second-order Trotter circuits, as we find analogous results for the other Trotter orders. For the chain we take compressed circuits with $M = 4, 8$, in which case the second-order Trotter circuits with similar optimized infidelity have $M = 5, 16$. Imporantly, while these compressed and Trotter circuits have similar fidelity, the $M = 5$ Trotter circuit has 1.4 times the gate count of the $M = 4$ compressed circuit, whereas the $M = 16$ Trotter circuit has 2.1 times the gate count of the $M = 8$ compressed circuit. For the ladder we take compressed circuits with $M = 8, 16$, such that the corresponding second-order Trotter circuits have $M = 2, 4$, i.e. they contain 1.6 times as many gates.

The results are displayed in Fig. 6 in log-log scale, where in the left panel we show the stacked infidelities for the chain and in the right panel for the ladder. The red dashed lines are for the power laws $\epsilon \propto t^n$ with the best fitting power $n$. It is seen that the infidelity increases similarly for all considered pairs of compressed and Trotter circuits, which like Fig. 5 emphasizes that the compression strategy expounded in Sec. 2 has no drawbacks at long times, relative to the Trotter circuits. Moreover, the mentioned discrepancy in gate counts, with in all cases the Trotter circuit having significantly more gates, makes the compressed

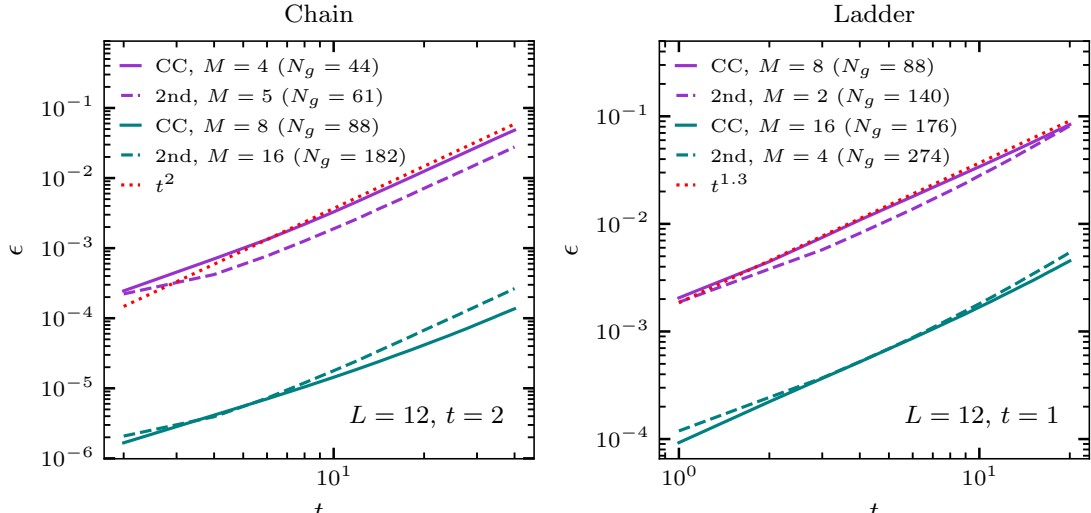

Figure 6: The infidelity $\epsilon$ versus stacking time $t$ for the time evolution operator of the $L = 12$ Heisenberg model on a chain at $t = 2$ (left panels) and ladder at $t = 1$ (right panels), for compressed and second-order circuits that are stacked twenty times. The circuits were chosen such they have similar $\epsilon$ at the optimized $t$, with $M = 4, 8$ and $M = 5, 16$ for compressed and second-order Trotter circuits on the chain, and $M = 8, 16$ and $M = 2, 4$ for the ladder. As a result the compressed circuits have significantly lower gate count than the corresponding Trotter circuits. The red dashed lines are for the power laws $\epsilon \propto t^n$ with the best fitting power $n$.

circuits especially favorable for simulation on real quantum devices, where the error due to gate imperfections and decoherence noise hampers time evolution.

## 3.2 Out-of-time-ordered correlators

Having studied the infidelity and its behavior under stacking in detail in Sec. 3.1, we now use the compressed circuits to determine the behavior of a quantity that does not enter the objective function (19), namely the OTOC (22).

In Fig. 7 we show the absolute $C_{i=2,j}(t)$ errors, relative to the targeted time-evolution operator, for compressed circuits which were optimized for $L = 8, 12, 16$ chains and ladders at $t = 2$ and stacked up to ten times, along with the errors for Trotter circuits with gate counts similar to these compressed circuits. For the chain we let $j$ run over all sites, whereas for the ladder it runs over all rungs. The upper three rows are for the chain while the lower three rows are for the ladder. The first and fourth row are for $L = 8$, the second and fifth row are for $L = 12$, and the third and sixth row are for $L = 16$. The left column is for the compressed circuit while the second, third and fourth columns are for the first-, second- and fourth-order Trotter circuits. As in Fig. 5 the depths are $M = 8, 8, 7, 1$ for the chain and $M = 16, 4, 3, 1$ for the ladder, for the compressed circuit and first-, second- and fourth-order Trotter circuits, respectively.

For the chain it is clear that the compressed circuit works better than the Trotter circuits within the lightcone, whereas it is slightly worse than the second- and fourth-order Trotter circuits at approximating the small values outside of the lightcone. For the ladder the compressed circuit is better everywhere, even better than the fourth-order Trotter circuit which has twice as many gates. Hence we draw the same conclusion as from Fig. 5: With a similar amount of gates we are able to go farther in time with the compressed circuits than with the

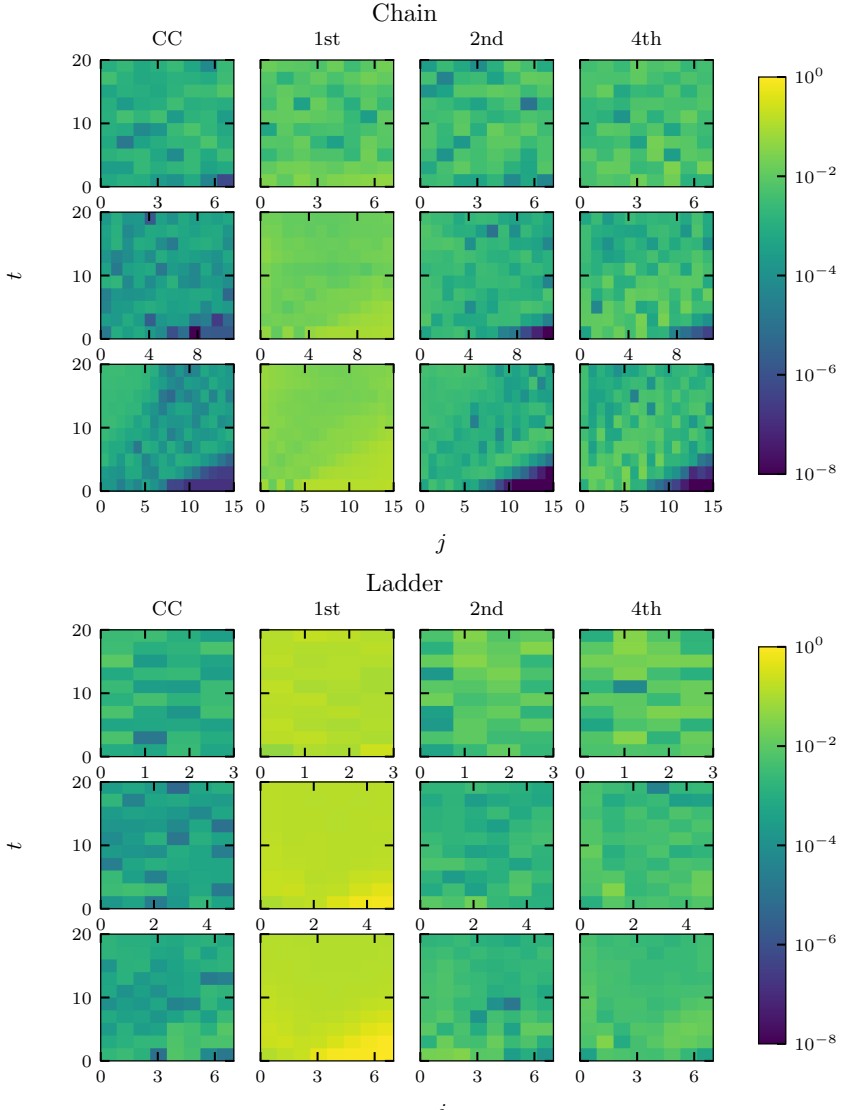

Figure 7: The absolute $C_{i=2,j}(t)$ errors for the chain (top three rows) and ladder (bottom three rows) for a compressed circuit optimized at $t = 2$ and stacked up to ten times, along with the errors for Trotter circuits with similar gate counts. For the chain $j$ labels the sites and for the ladder it labels the rungs. The first and fourth row are for $L = 8$ with $\chi = 256$, the second and fifth row are for $L = 12$ with $\chi = 150$, and the third and sixth row are for $L = 16$ with $\chi = 100$. The first column is for the compressed circuit, the second, third and fourth columns are for the first-, second- and fourth-order Trotter circuits. To have roughly equal gate counts, the used depths are $M = 8, 8, 7, 1$ for the chain and $M = 16, 4, 3, 1$ for the ladder, for the compressed circuit and first-, second- and fourth-order Trotter circuits, respectively.

Trotter circuits, before reaching some error threshold, even though we do not optimize based on OTOCs.

In Sec. C we show the OTOC values corresponding to the errors from Fig. 7, for compressed circuits and the targeted time-evolution operators. There we also show how the relative error of $C_{i=2,j=4}(t)$ propagates with stacking, for compressed and Trotter circuits that have similar optimized fidelity, indicating that we can maintain similar fidelity with compressed circuits that have a fraction of the amount of gates of the Trotter circuits.

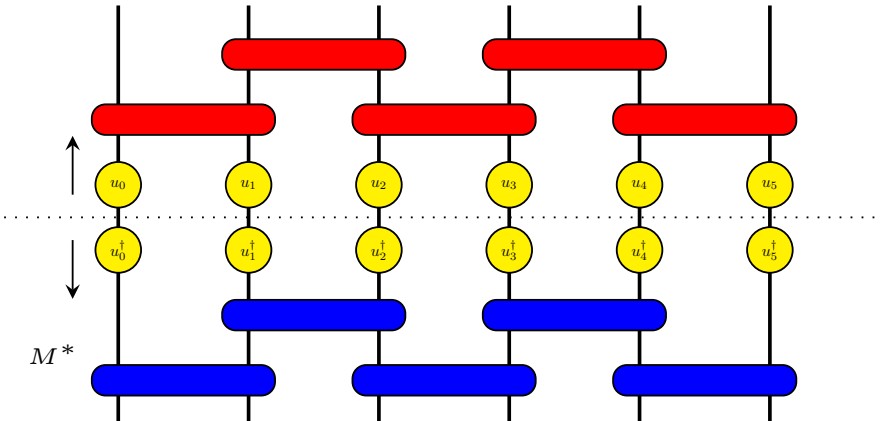

Figure 8: The gauge freedom that exists between the layers of a circuit. When we cut the circuit across the horizontal dashed line, and want to use the lowest $M^*$ layers to calculate an infidelity, we have to take into account the gauge freedom that is encoded by inserting a pair of conjugate one-qubit unitaries $u_i^\dagger u_i = I$ at each qubit, and absorbing one unitary upwards and the other downwards.

## 3.3 Analysis of the compressed circuit

In the previous Sections 3.1 and 3.2 we have seen that the compressed circuit outperforms the Trotter circuits. Here we investigate how this is achieved, by probing the structure of the layers and gates that make up the compressed and Trotter circuits.

Starting off, we take a compressed circuit and Trotter circuits with comparable gate counts, and consider the infidelity between a subset of layers $M^* < M$ (counting from the bottom layer) and the time evolution operator at a time $t^* < t$ that is smaller than the time-step $t$ at which the compressed circuit was optimized. Crucially, we must take into account the gauge freedom that exists between layers, where we are able to insert conjugate layers of one-qubit unitaries, and absorb one layer into the subset we are considering and the other layer into its complement. This process is illustrated in Fig. 8. Hence when calculating a subset infidelity for the compressed circuit, we add a layer of one-qubit unitaries between the subset and the time evolution operator at $t^*$, and minimize the infidelity with respect to these one-qubit unitaries. This way we account for the gauge freedom.

In Fig. 9 we show the results for the chain with $L = 8$ at $t = 1$, for a compressed circuit with $M = 8$ and Trotter circuits with $M = 8, 7, 1$ for first-, second- and fourth-order, which have gate counts close to that of the compressed circuit. Here we define a Trotter circuit with $M^*$ layers as having $M^*$ brickwall layers, and the largest shown $M^*$ is the full circuit, which e.g. for the second-order Trotter circuit involves adding half a brickwall layer to its largest subset. For the compressed circuit $M^* = 8$ corresponds to the full circuit. The dashed lines mark the times $t^* = tM^*/8$.

From Fig. 9 it is clear that at $t = 1$ there is significant overlap of the subsets with a time evolution operator at $t^* < t$ for both the compressed and Trotter circuits. However, in contrast to the first- and second-order Trotter circuits, where the infidelity dips are equidistant, and where for the first-order Trotter circuit the dip depth is decreasing with the number of stacked layers while for the second-order Trotter circuit it is constant, the dips of the compressed circuit are instead roughly symmetric and are smallest around $t^* \approx t/2$. A closer look reveals that

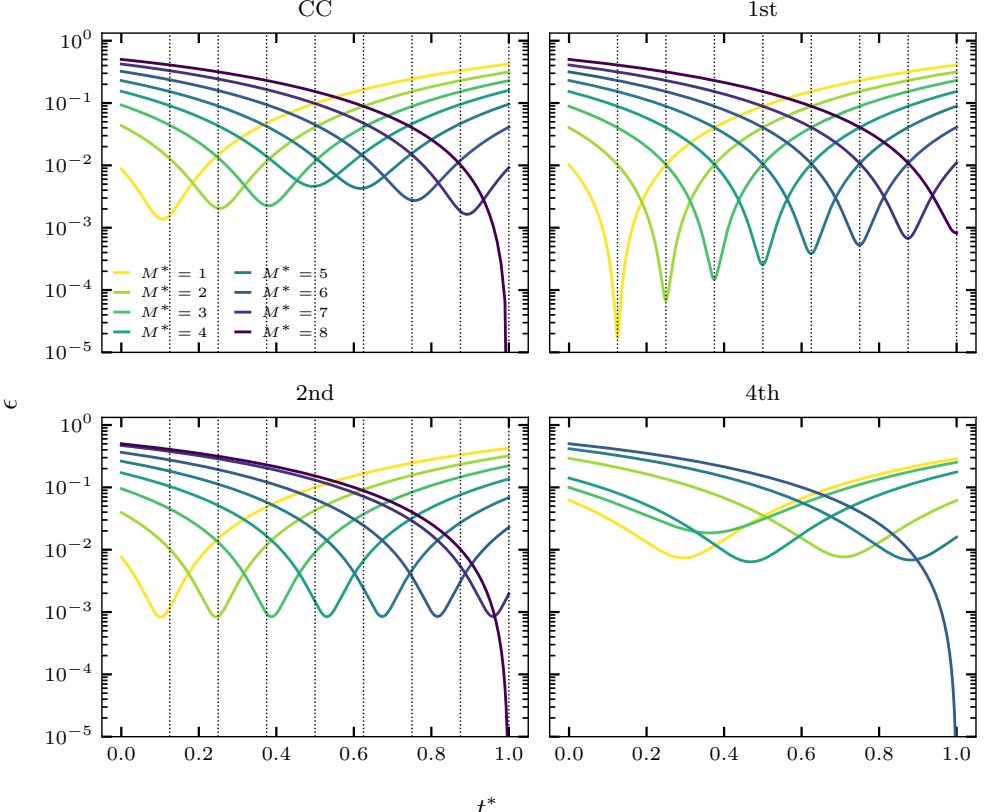

Figure 9: The infidelity between a subset of layers $M^* < M$, counting from the bottom layer, and the targeted time evolution operator at time $t^* < t$, where $t$ denotes the time-step at which the compressed circuit was optimized. The plots are for a Heisenberg chain with $L = 8$ at $t = 1$. In the top left panel we show the results for a compressed circuit with $M = 8$, in the top right for a first-order Trotter circuit with $M = 8$, in the bottom left for a second-order Trotter circuit with $M = 7$, and in the bottom right for a fourth-order Trotter circuit with $M = 1$. These depths were chosen such that the circuits have similar gate count. The curve with $M^* = M$ corresponds to the full circuit. The dashed lines mark times $t M^*/8$.

the infidelity at this point is roughly $10^{-2}$, which is more than one order of magnitude larger than for the first- and second-order Trotter circuit at similar $t^*$. This is even more remarkable when taking the final infidelity into account, which is $\epsilon = 1.8 \cdot 10^{-9}$ for the compressed circuit and therefore at least three orders of magnitudes better than the first-, second- and fourth-order Trotter circuits, which have $\epsilon = 8.2 \cdot 10^{-4}, 1.2 \cdot 10^{-6}, 2.1 \cdot 10^{-6}$. This indicates that the compressed circuit does not follow the target "trajectory" given by the unitary time evolution, but slightly deviates from it. However, it becomes "refocused" at $t^* = t$, which we sketch in Fig. 10. It is an interesting question for future research to understand the alternative trajectory, which might be beneficial for an optimal discretization of time evolution beyond the Trotter decomposition. In Sec. C we show that the refocussing also occurs for the OTOCs.

We note that we did not find these symmetric dips for all our compressed circuits, especially for larger $t$ and the ladder geometry. It remains an open question whether this is an artefact of the convergence of the optimization to a non-global minimum.

As a further comparison between compressed and Trotter circuits, we calculate the operator entanglement entropy (opEE) of their gates [47, 56]. Concretely, we take an optimized compressed circuit $\mathcal{C}$ and decompose each two-qubit gate $U_{ij} \in \mathcal{C}$ using a singular value de-

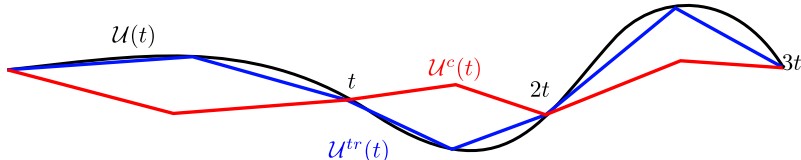

Figure 10: A sketch of the "refocussing" mechanism that potentially explains the structures observed in Fig. 9. Here the targeted time evolution $\mathcal{U}(t)$ is shown in black, the Trotter evolution $\mathcal{U}^{tr}(t)$ is shown in blue, and the compressed evolution $\mathcal{U}^{c}(t)$ is shown in red. While $\mathcal{U}^{tr}(t)$ follows the target trajectory quite closely, $\mathcal{U}^{c}(t)$ instead becomes "refocussed" at multiples of the optimization timestep $t$.

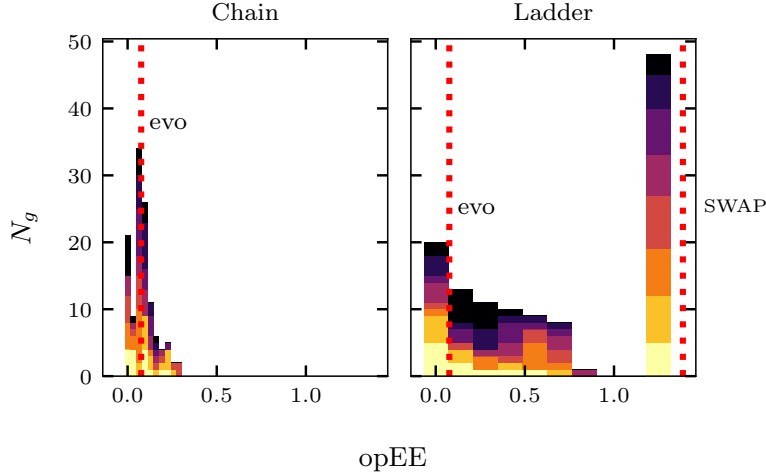

Figure 11: Stacked histograms for the opEE of the gates of a compressed circuit with depth $M = 8$, optimized at $t = 2$ for a $L = 16$ chain (left panels) and ladder (right panels). The colors denote the contents of each layer, with the lightest color for the bottom layer and the darkest for the top layer. The red vertical lines denote the values for the gates in a $M = 8$ first-order Trotter circuit, with the two lines in the ladder plots corresponding to the evolution and SWAP gates.

composition into

$$U_{ij} = \sum_{l=1}^{4} s_l v_i^l \otimes v_j^l, \tag{24}$$

where $v_i^l$ and $v_j^l$ are two sets of four one-qubit operators, acting on qubit $i$ and $j$ respectively, and where the four singular values $s_l$ encode the opEE of $U_{ij}$ as

$$\text{opEE} = -\sum_l s_l^2 \ln(s_l^2). \tag{25}$$

In Fig. 11 we display the opEE of all gates in a $M = 8$ compressed circuit for the chain (left panel) and ladder (right panel) for $L = 16$ at $t = 2$. The histograms are stacked, with each color denoting the content of a layer, where the lightest color represents the bottom layer and the darkest color the top layer. The red vertical lines mark the values for the $M = 8$ first-order Trotter circuit, with the two lines in the ladder plots corresponding to the evolution and SWAP gates. These histograms show that the gates of the compressed circuit are more hetergenous compared to those of the Trotter circuits, since they have a relatively large spread in opEE instead of one or two values. Moreover, for the ladder it is seen that a several gates in the

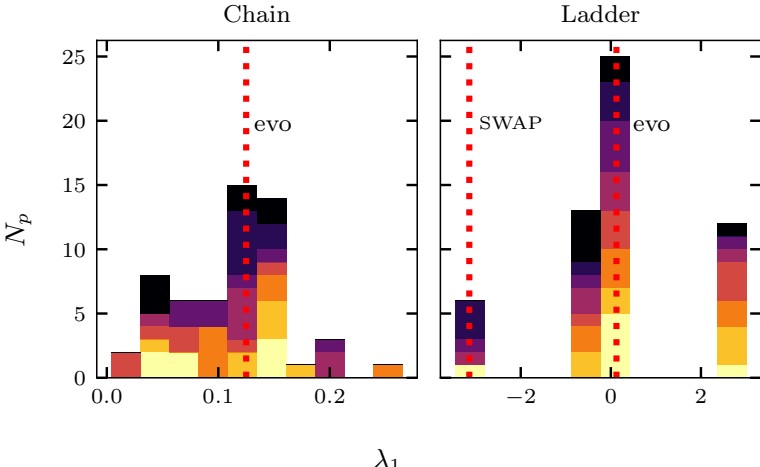

Figure 12: The distribution of the $\lambda_1$ parameter which enters the two-qubit unitary parameterization that was used in this work, shown for the chain (left panel) and ladder (right panel) with $L = 8$ at $t = 1$. The parameter count $N_p$ for a compressed circuit with $M = 8$ is shown as a stacked histogram, with the lightest color corresponding to the bottom layer and the darkest color to the top layer. The first-order Trotter evolution gate value $\lambda_1^{\text{evo}} = t/M$ and the SWAP gate value $\lambda_1^{\text{SWAP}} = -\pi$ are shown as dashed red lines. The other two-qubit parameters $\lambda_2$ and $\lambda_3$ are distributed similarly. Note the different scales of the x-axes.

compressed circuit assume an opEE that is near to that of the SWAP gate, which we view as an indication that the action of the SWAP gate is baked into our optimized circuits.

Finally we consider the distribution of the parameter $\lambda_1$ across the optimized two-qubit unitaries, which are parameterized as in (14). We found that $\lambda_2$ and $\lambda_3$ are distributed similarly. In Fig. 12 we show histograms for the parameter counts $N_p$ of $\lambda_1$ for the chain (left panel) and ladder (right panel) with $L = 8$ at $t = 1$, for a compressed circuit with $M = 8$. Note here the different scales of the x-axes. The histograms are again stacked, with the lightest color corresponding to the bottom layer and the darkest color to the top layer. The red dashed lines mark the values of the gates in the $M = 8$ first-order Trotter circuit, for which $\lambda_1^{\text{SWAP}} = -\pi$ and $\lambda_1^{\text{evo}} = t/M$, both having no one-qubit dressing (15). As in Fig. 11, we see that the gates of the compressed circuit have a larger spread than the gates of the Trotter circuit, which instead assume one or two values. Also, for the ladder we again observe an accumulation of gates near the SWAP value.

The gates appearing in the optimized circuits appear to encode more structure than gates from Trotter circuits and are generally speaking encoding a larger change of the wave function per gate compared to the case of Trotter circuits. This can be seen best in the limit of very small Trotter time steps, in which each appearing gate (except SWAP) is very close to identity, while in the opposite limit which we optimize for, each gate needs to be sufficiently different from identity in order to represent the same time evolution operator.

## 4 Conclusion and Outlook

In this work we have presented an approach which reduces the resource cost of digital quantum simulation compared to standard Trotter decompositions by globally optimizing a simple parameterized brickwall circuit in a way that is scalable to large systems. Crucially, the perfor-

mance per gate is better even when the compressed circuit does not respect the connectivity of the simulated lattice, potentially allowing for high fidelity simulation of systems with a connectivity that is larger than that of the used quantum processor. To illustrate this we have compared the infidelity of the compressed and Trotter circuits with the targeted time evolution operators of Heisenberg chains and ladders, as well as the ability to reproduce their OTOCs.

We have shown that we can achieve similar accuracy of the time evolution operator with up to one order of magnitude less gates, depending on the desired accuracy and system. Moreover, we checked that this advantage persists when stacking the circuits many times, a central ingredient to simulating a quantum system over long times. This enables high fidelity propagation to times which are currently elusive with conventional Trotter decomposition methods.

Furthermore, we analyzed the structure of the compressed circuits. In the case of the chain, we observed a "refocussing" mechanism, which suppresses the infidelity at multiples of the optimized time step, while the evolution inside the optimized circuit appears to follow a trajectory which is further away from the targeted time evolution operator. It is an interesting question for further research to understand this trajectory and relate it also to recent studies of Trotter decompositions and its breakdown for large time steps [34, 35].

Our results open the door for many further directions. As a next step, one can for example take symmetries into account to further reduce the number of parameters. This might be especially favorable when exploiting translation symmetries. Furthermore, one can optimize the circuits with other cost functions than the fidelity, as was also done for example in [37]. Promising directions are using local observables or density matrices. While such an approach might simplify the convergence of the optimization, it is still an open question to what extent the accurate simulation of observables or other general quantities would be recovered.

We end by stressing that in this work we have used the simplest possible noise model, by assuming that each applied gate introduces the same amount of noise to the system and that therefore a minimization of the gate count reduces the overall noise. A refinement of this noise model will be the subject of future research.

## Acknowledgments

We thank Luis Colmenarez for useful comments on the manuscript. D.H. thanks Adam Smith, Frank Pollmann, and Hongzheng Zhao for useful discussions.

**Funding information** This project was supported by the Deutsche Forschungsgemeinschaft (DFG) through SFB 1143 (project-id 247310070) and the cluster of excellence ML4Q (EXC 2004, project-id 390534769). We also acknowledge support from the QuantERA II Programme that has received funding from the European Union's Horizon 2020 research innovation programme (GA 101017733), and from the Deutsche Forschungsgemeinschaft through the project DQUANT (project-id 499347025).

## A  Gate count equations

Here we state the equations for the NN two-qubit gate counts $N_g$ of the first-, second- and fourth-order Trotter circuits of depth $M$, which are used in Sec. 3. These are denoted by $N_{g\mathfrak{c}/\mathfrak{l}}^{1\text{st}}(M)$, $N_{g\mathfrak{c}/\mathfrak{l}}^{2\text{nd}}(M)$ and $N_{g\mathfrak{c}/\mathfrak{l}}^{4\text{th}}(M)$, respectively, where $\mathfrak{c}$ corresponds to the chain and $\mathfrak{l}$ to the triangular ladder. In deriving these equations we made maximal use of the ability to combine gates in subsequent Trotter steps. The compressed circuits have gate count $N_{g\mathfrak{c}}^{1\text{st}}$.

For the chain the equations are

$$N_{g\mathfrak{c}}^{\text{1st}}(M) = M(L-1), \tag{A.1}$$

$$N_{g\mathfrak{c}}^{\text{2nd}}(M) = M(L-1) + \left\lfloor \frac{L}{2} \right\rfloor, \tag{A.2}$$

$$N_{g\mathfrak{c}}^{\text{4th}}(M) = 5M(L-1) + \left\lfloor \frac{L}{2} \right\rfloor. \tag{A.3}$$

For the ladder, in which case we have to take into account the SWAP gates, the corresponding equations are

$$N_{g\mathfrak{l}}^{\text{1st}}(M) = M(4L-7), \tag{A.4}$$

$$N_{g\mathfrak{l}}^{\text{2nd}}(M) = 2M\left(N_{g\mathfrak{l}}^{\text{1st}}(1)+1\right) - (3M-1)\left\lfloor \frac{L}{2} \right\rfloor, \tag{A.5}$$

$$N_{g\mathfrak{l}}^{\text{4th}}(M) = 5MN_{g\mathfrak{l}}^{\text{2nd}}(1) - (5M-1)\left\lfloor \frac{L}{2} \right\rfloor. \tag{A.6}$$

## B  Convergence of the optimization

In order to find the optimal compressed circuit using the gradient descent method outlined in Sec. 2.3, it is important to scan the hyperparameter space of the used optimizer. The reason is that there is no single set of hyperparameters which finds the best solution for all optimization problems. We find the best convergence by using the vanilla Adam optimizer [52], which is presented in Algorithm 1.

We scan the hyperparameter space $(\lambda, \delta, \beta_1, \beta_2)$ for the most favorable convergence properties. As mentioned in Sec. 2.3, it is crucial to continue iterating the algorithm when we

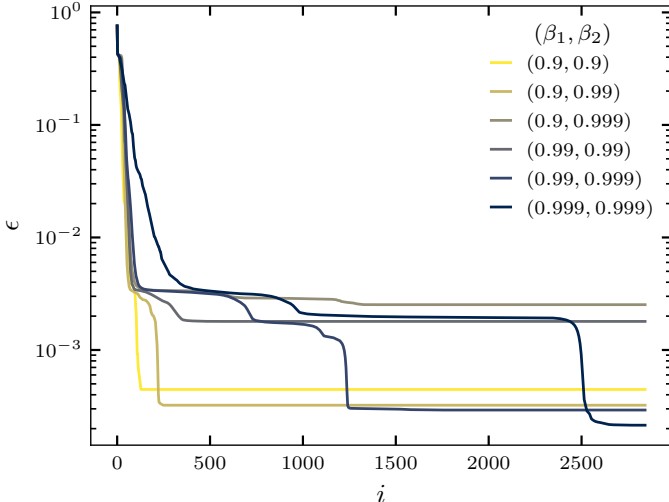

Figure 13: The infidelity $\epsilon$ as a function of the iteration step $i$ for an Adam optimizer with learning-rate $\tau = 0.01$, regularization $\delta = 10^{-4}$, and various decay rates $(\beta_1, \beta_2)$ with $\beta_1, \beta_2 \in \{0.9, 0.99, 0.999\}$. The optimization is performed for a size $L = 8$ ladder at time $t = 1$ with circuit depth $M = 8$. The lowest infidelity is reached with $(0.999, 0.999)$, but crucially this requires the optimizer to spend time in local minima without getting stopped by a convergence criterion when the infidelity has stagnated.

---

**Algorithm 1** *Vanilla Adam [52].* This gradient-descent optimizer updates the circuit parameters $\vec{\theta}$ to minimize the infidelity $\epsilon(\vec{\theta})$, by taking into account exponentially decaying running averages of the first moment $m$ and second moment $v$ of the infidelity gradient $g$ for each parameter separately. Instead of choosing the parameter updates to be proportional to $g$, as in vanilla gradient descent, here it is proportional to a memory of the previous gradients $m$. This results in a relatively stable minimization and to some extent prevents getting stuck in local minima. Moreover, since the optimization algorithm is first order, the magnitude of the parameter update is proportional to its uncertainty in decreasing the infidelity. For this reason, large updates are undesirable, whereas tiny updates are also undesirable since they halt the minimization and promote getting stuck in local minima. With this in mind, the update magnitude is forced to be desirable, by choosing it to be proportional to $m/\sqrt{v}$.

---

**Hyperparameters:**
  $\lambda$: Raw learning-rate
  $\beta_1$: First moment decay strength
  $\beta_2$: Second moment decay strength
  $\delta$: Regularization
  $N_{\text{iters}}$: Amount of iterations
**Initial conditions:**
  $m_0 \leftarrow 0$ (First moment initially zero)
  $v_0 \leftarrow 0$ (Second moment initially zero)
**for** ( $i = 0$; $i < N_{\text{iters}}$; i = i+1 ) **do**
  $g_i \leftarrow \nabla_{\vec{\theta}_{i-1}} \epsilon(\vec{\theta}_{i-1})$ (Calculate gradient at current parameters)
  $m_i \leftarrow \beta_1 m_{i-1} + (1-\beta_1)g_i$ (Extend running average of first moment)
  $m_i^* \leftarrow m_i/(1-\beta_1^i)$ (Bias correction)
  $v_i \leftarrow \beta_2 v_{i-1} + (1-\beta_2)g_i^2$ (Extend running average of second moment)
  $v_i^* \leftarrow v_i/(1-\beta_2^i)$ (Bias correction)
  $\vec{\theta}_i \leftarrow \vec{\theta}_{i-1} - \lambda m_i^*/(\sqrt{v_i^*} + \delta)$ (Update parameters)
**end for**
**return** $\vec{\theta}_i$ (Final circuit parameters)

---

reach a plateau in the fidelity. This is illustrated in Fig. 13, where we display the gradient descent of $\epsilon$ for a circuit with $M = 8$ layers on the time evolution operator of an $L = 8$ ladder at $t = 1$, and consider various $(\beta_1, \beta_2)$ with learning-rate $\tau = 0.01$ and regularization $\delta = 10^{-4}$. Here the largest fidelity is obtained with $\beta_1 = \beta_2 = 0.999$, but we have to overcome multiple plateaus, which would have been spoiled by using a convergence criterion.

## C  OTOC details

First we display the OTOC values of the stacked compressed circuits and targeted time-evolution operators that were used to make Fig. 7. In the left two panel columns of Fig. 14 we show the OTOCs $C_{i=2,j}(t)$ for the chain and in the right two columns for the ladder. The first and third columns are for the compressed circuits, whereas the second and fourth columns are for the target unitaries. As already became apparent from Fig. 7, the agreement is excellent for all considered stacking times $t$.

Now we consider the analog of Fig. 6 for the relative error of the OTOC $C_{i=2,j=4}$. In particular, we consider the chain and ladder with $L = 12$ and take a couple compressed circuits for which the infidelities were optimized at $t = 2$ for the chain and $t = 1$ for the ladder, which we then stack up to $t = 20$. As in Fig. 6 we take compressed circuits with $M = 4, 8$ for the

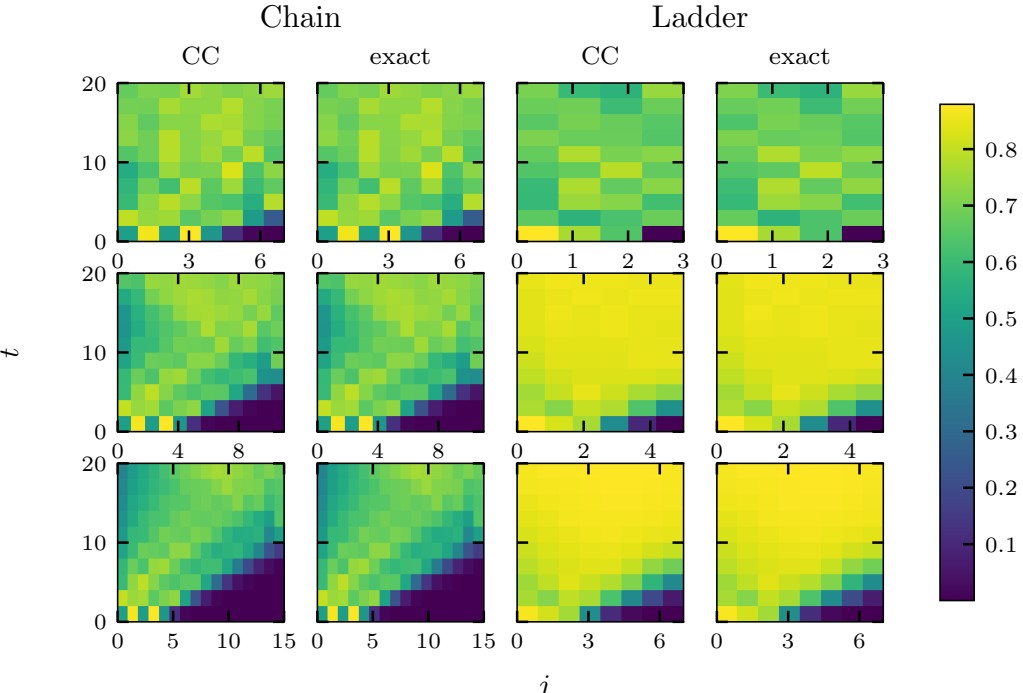

Figure 14: The OTOCs $C_{i=2,j}(t)$ as a function of site or rung $j$ and stacking time $t$ for the chain (left columns) and the ladder (right columns), for compressed circuits optimized at $t = 2$ and stacked up to ten times (first and third columns) and the corresponding target values (second and fourth columns). For the chain we take $M = 8$ and for the ladder $M = 16$. The top row is for $L = 8$ with $\chi = 256$, the middle row is for $L = 12$ with $\chi = 150$, and the bottom row is for $L = 16$ with $\chi = 100$.

chain and $M = 8, 16$ for the ladder, and we compare these with second-order Trotter circuits that have similar fidelity at the optimized time step, corresponding to $M = 5, 16$ for the chain and $M = 2, 4$ for the ladder. In Fig. 15 we show the results, with the left panel for the chain and the right panel for the ladder. The implications are the same as those derived from Fig. 6: With a smaller amount of gates we essentially get the same performance, in this case even for a quantity that does not appear in the objective function (19).

Finally, we check whether the refocussing that was observed for the infidelity $\epsilon$ in Fig. 9 also emerges for the OTOCs, which contrary to $\epsilon$ does not enter the cost function of the optimization scheme. In Fig. 16 we show the relative error of $C_{i=5,j=5}(t^*)$ between that of $M^*$ layers and that of the target unitary at time $t^*$. Before using the $M^*$ layers to calculate the OTOC at $t^*$, we minimize its infidelity with respect to the target unitary at $t^*$, taking into account the gauge invariance. As in Fig. 9, we perform the calculations for the Heisenberg chain with $L = 8$ and a $M = 8$ circuit optimized at $t = 1$, with the results shown in Fig. 16. We see that a similar refocussing takes place, with the minima for $M^* < M$ being elevated with respect to that at $M^* = M$ and with unequal spacing in time.

## D  Emergence of lattice symmetries

The brickwall circuit ansatz (18) used in this work has the most general form, consisting of arbitrary two-body unitaries and not taking into account any symmetry of the targeted time-evolution operator, i.e. in our case those corresponding to the Hamiltonians (2). To restrict

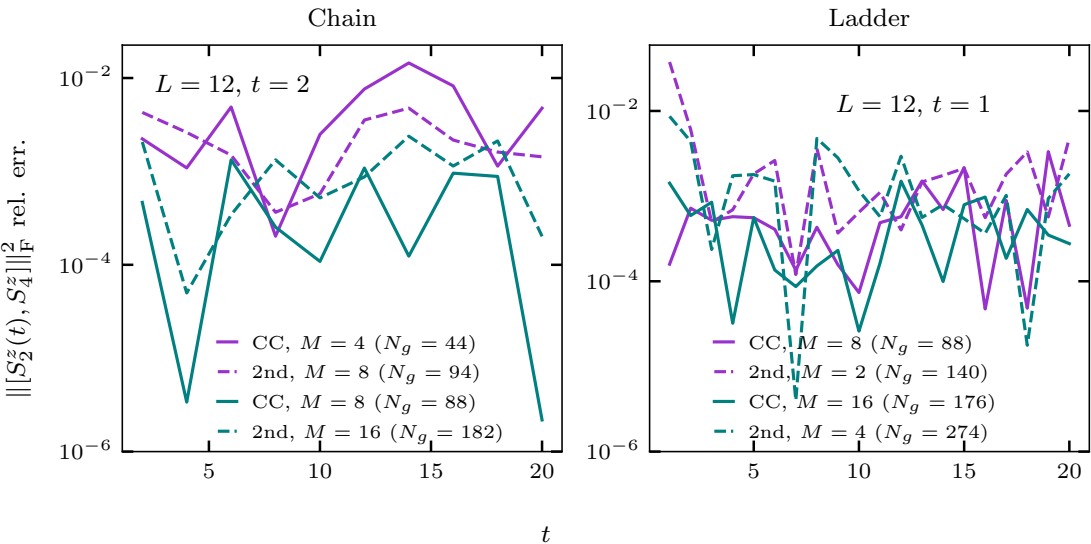

Figure 15: The relative error of the OTOC $C_{i=2,j=4}(t)$ versus stacking time $t$ for the chain (left panel) and ladder (right panel) with $L = 12$, for circuits optimized at $t = 2$ for the chain and $t = 1$ for the ladder, and stacked up to time $t = 20$. For the chain we consider $M = 4, 8$ and for the ladder $M = 8, 16$. For each $M$ we choose a second-order Trotter circuit with similar fidelity at the optimized $t$, i.e. $M = 5, 16$ for the chain and $M = 2, 4$ for the ladder. As a result the compressed circuits have significantly less gates than the corresponding Trotter circuits.

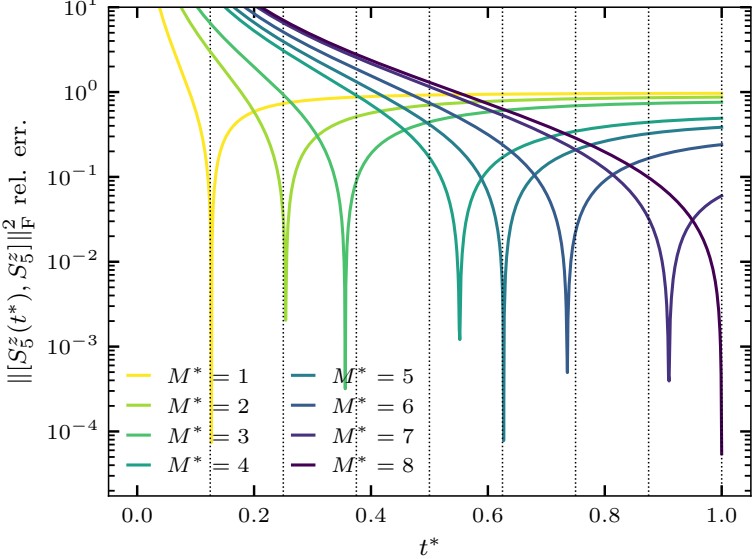

Figure 16: The relative error of the OTOC $C_{i=5,j=5}(t^*)$ between that of a subset $M^*$ of layers and that of the target unitary at time $t^*$, for the Heisenberg chain at $L = 8$ and a depth $M = 8$ circuit optimized at time $t = 1$. The curve with $M^* = M$ corresponds to the full circuit and the dashed lines mark times $tM^*/8$.

the ansatz space it could be useful to incorporate such symmetries into the circuit at the gate level.

Take for example the Heisenberg chain in (2), which possesses lattice inversion symmetry,

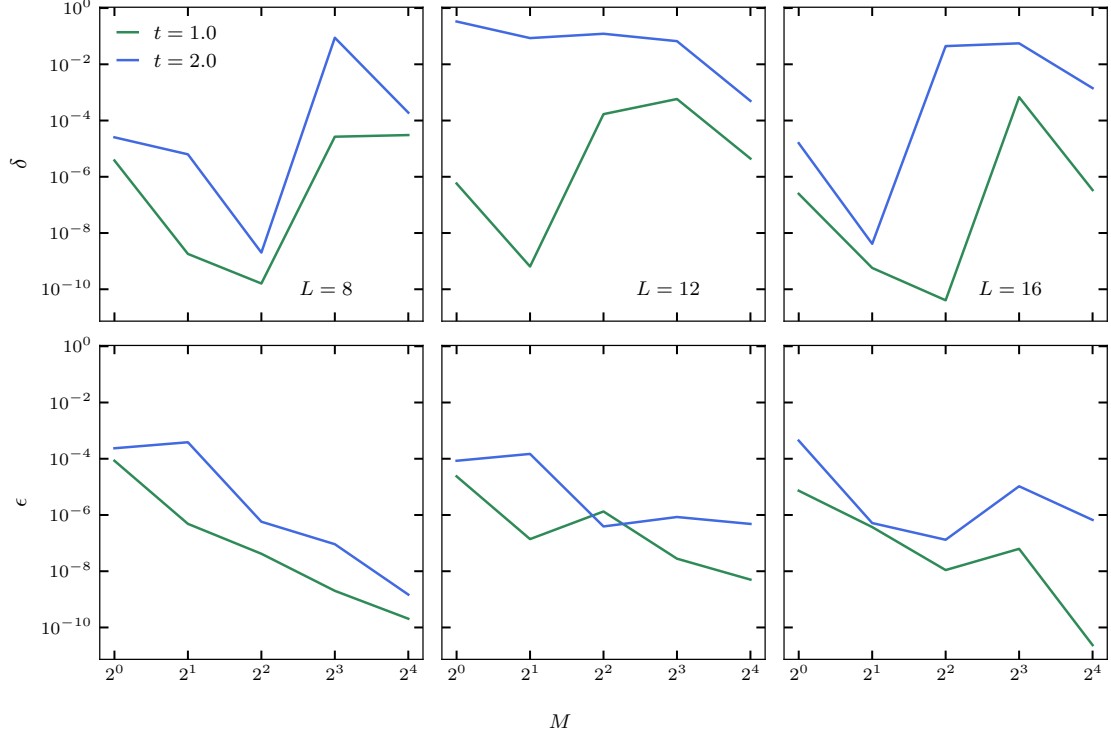

Figure 17: The average gate-wise infidelity $\delta$ of every gate with its mirrored counterpart (top panels), flipped across the middle bond, for all gates in compressed circuits which were optimized to approximate the lattice inversion symmetric Heisenberg chain time-evolution operator. For comparison, we also show the infidelity $\epsilon$ of the circuit as a whole with its mirrored counterpart (bottom panels). These quantities probe to which extent the inversion symmetry of the targeted unitary emerges in the compressed circuit. The infidelities are shown as a function of the circuit depth $M$, for times $t = 1$ and $t = 2$. The left panels are for system size $L = 8$, the middle panels are for $L = 12$, and the right panels are for $L = 16$.

being invariant under a flip of the lattice across the middle bond for even $L$. To incorporate this into the ansatz we let the gate acting on the bond between sites $i$ and $i + 1$ also act on the mirrored bond between $L-2-i$ and $L-1-i$, albeit flipped across the time axis. Since this gate and its flipped counterpart should be equal for the inversion symmetry to be manifest, the gate parameterization (13) implies that the one-qubit unitary $u_i$ should be equal to $u_j$, and that $u'_i$ should be equal to $u'_j$, with the two-qubit unitary $v_{ij}$ being flip-symmetric by construction.

Since we did not incorporate this inversion symmetry into the circuits used for our simulations, it is an interesting question whether the chosen circuit ansatz in combination with the optimization procedure leads to its emergence. To probe this, we take an optimized circuit and for each of its gates we calculate the infidelity with its mirrored counterpart, and then average over all gates to get the average gate infidelity $\delta$. As for the subset infidelity from Fig. 9, here it is crucial to take into account the gauge symmetry. We also calculate the infidelity $\epsilon$ of the circuit as a whole with its mirrored counterpart, to determine if it is reasonable to expect the symmetry to emerge on the gate level. If this overall infidelity is high, it is unlikely that it is low at the gate level. The results are shown in Fig. 17.

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
