# Peer review of "Optimal compression of quantum many-body time evolution operators into brickwall circuits"

_SciPost Physics, doi:SciPost Phys. 14, 073 (2023)_

## Round 1 · Referee Report · Subhayan Sahu (Referee 1) · 2022-7-4

Report

In this article, Tepaske et al study methods to obtain optimal brickwork circuit for time evolution operators generated by 1-d local spin Hamiltonians. They optimize the brickwork circuit with respect to an objective function defined as the Frobenius distance between the exact time evolution operator and the brickwork unitary, and also compare the gate count and fidelity with respect to the various orders of Trotter evolution. Apart from the distance from the exact unitary, they also compute OTOCs of local operators, and show that the OTOCs are efficiently captured by the optimized circuit.

The results and analysis reported in the article are sound, and overall appear to be significant, especially in the context of simulating time evolution on quantum simulation platforms which are limited by geometric locality. I do have a few questions and suggestions for the authors, and I think the paper warrants a publication in this journal following their response and modifications.

  1. The authors don't provide much detail in the manuscript about the optimization process, especially how much resource and optimization time is required for achieving the quoted results. This is relevant, since that will determine the scalability and accuracy of the optimization procedure. The authors comment that they encounter the 'barren plateau' in optimization, but do not provide any quantitative details. It would be useful if the authors could provide more details of the parameters and the performance of the optimization procedure, and also comment on the scalability of the optimization to larger systems, at least in an appendix. It would also be useful to visualize the `slow' convergence that they comment upon. It would also help the reader to briefly explain the differential programming method that they employ.

  2. In the discussions, the authors comment upon the scope of using their methods with translational invariance. It would be interesting to check and present details about whether the optimized gates show any emergent translational invariance, since the underlying dynamics that they consider is translationally invariant. Do the results change significantly if the gates are enforced to be uniform (when periodic boundary conditions are considered)?

  3. Regarding the OTOCs, one suggestion is to improve the presentation of the data in the main text, by showing the error in the optimized/Trotter circuits with respect to the exact computation in the space-time diagrams, since it is difficult to compare the different panels in Fig. 7. It would probably be better to show Fig. 14 in the main-text as that is a better comparison of the method with the exact computation. Furthermore, it would be interesting to compare the intermediate-time OTOCs for an optimized circuit, to explore the refocusing phenomenon that the authors found in some numerical experiments (in the discussion around Fig. 9 and 10). Do the OTOC data show an elevated error at intermediate times, which becomes low again at late times?

  4. On a minor note, there are some comments (second paragraph, pg 10) about how the performance of the compressed optimized circuits has similar asymptotic slopes with the best Trotter circuits in Fig. 4. However, the data in the figures show that the 4th order Trotter circuits have consistently more favorable slopes for the parameters considered in the plot. It would be better to qualify or clarify that comment.

  • validity: high
  • significance: good
  • originality: good
  • clarity: good
  • formatting: excellent
  • grammar: excellent

Author:  Maurits Tepaske  on 2022-10-24  [id 2949]

(in reply to Report 1 by Subhayan Sahu on 2022-07-04)

First of all, thank you for your extensive report. Below we reply to your comments.

  1. We agree with these statements. We have added an appendix with more details on the optimization algorithm and some visualization on how the cost function behaves during the optimization to emphasize the mentioned convergence behavior.
  2. This is indeed an interesting point. So far, we have not yet investigated the case of translation invariance, since the algorithm changes considerably if we enforce periodic boundary conditions or consider infinite systems. We did however check that the optimized gates indeed respect the lattice symmetries of the Heisenberg model with open boundaries, i.e. lattice inversion around the central bond. We have added an appendix where we discuss this case.
  3. We agree and have now put the errors in the main text. We have also extended the OTOC appendix with a plot for the OTOC refocussing.
  4. We also agree here, the plots do not agree with this statement for L=12 and L=16. We have modified the text to reflect this.

---

## Round 1 · Referee Report · Michael Flynn (Referee 2) · 2022-10-2

Strengths

  1. The authors do a nice job of introducing their work such that experts and non-experts alike can appreciate the significance of their results.

  2. The presentation is quite thorough and the authors clearly anticipate the questions their work will raise for readers.

Weaknesses

  1. The presentation can be difficult to follow in some places (see report).

Report

In this manuscript, the authors study numerical methods which optimally implement time-evolution operators of local one-dimensional spin models as brickwork circuits. By optimizing this circuit representation with respect to an objective function (the "infidelity", or normalized Frobenius distance between the time-evolution operator and bricklayer circuit), they argue that this representation encodes time evolution with greater fidelity for fixed resources than Trotter decompositions. They supply evidence in the form of direct comparisons between bricklayer and Trotter circuits, normalizing for resources such as the number of gates required to achieve a particular fidelity. Beyond the objective function, they also demonstrate that the bricklayer form can be used to efficiently compute observables such as out-of-time-order correlation functions of local operators.

I find the results and analysis of the article to be both sound and significant. The authors do a nice job of motivating this study in the context of digital quantum simulation, where engineering limitations demand that computations be implemented with minimal resources. Techniques such as those presented here are clearly valuable in the NISQ era. I believe this paper warrants publication in SciPost Physics, though I would ask the authors to address the following questions and comments, including revisions as necessary.

  1. In general, the authors have provided thorough details throughout the manuscript regarding parameters used in simulations, definitions, etc. However, this is currently lacking in the discussion of gradient descent, where the authors are quite vague when it comes to details of the optimization problem used to construct their bricklayer circuit. There is a particularly interesting comment left dangling about the "barren plateau" problem; these additional details seem interesting enough to experts to warrant an appendix, and would be particularly useful for others interested in conducting follow-up studies.

  2. The discussion of figure 5, particularly in the case of ladders, is misleading at the moment. It appears that the 4th order Trotter circuit outperforms the compiled circuit in some cases; the authors address this by mentioning this is not an "apples to apples" comparison because the Trotter case requires more gates. The discussion of this requires more clarification since precise counting of gates is relegated to an appendix. Explicitly enumerating the gate counts in these circuits would be nice, particularly since the authors tend to use the same reference circuits several times (e.g., the M=8,8,7,1 sequence for the chain).

  3. Another point worthy of clarification in the main text is the author's use of the word "exact". The authors state on page 9: "For all practical purposes, we will then consider this MPO our exact time evolution operator, which we want to approximate by our circuits." I assume this is the sense in which the word "exact" is used from that point onwards. However, in some contexts I suspect readers could be confused by this terminology. For example, figure 7 refers to "exact" OTOCs; for clarity, it would be nice to add a note about how this is computed.

  4. An important motivation of the authors is the application of their results to digital quantum simulation; as they discuss, it is important for this purpose that their bricklayer circuits serve as approximators for the dynamics of systems which are not 1D chains. More discussion of the role of geometry seems important in this regard: for example, in figure 6, different power law scalings are obtained for the chain and ladder geometries. Can these be related to the geometry in a neat way? More generally, one can imagine studying a quasi-1D geometry as is often encountered in tensor network studies. In such a system it seems that more swap gates, at least for local Trotter methods, would be necessary. Can the authors make any comment on how their method would perform in such a setting vs. Trotter circuits? This may be less relevant to direct applications in quantum simulation, but are worth commenting on so that the reader more fully understands the role of lattice geometry in this problem.

  5. The authors have already commented, in the concluding section, on the potential use of symmetries to reduce the number of free parameters in their optimization problem. Full-fledged work on this problem could lead to a manuscript in its own right, but additional commentary on the realization of symmetries in their circuit ansatz would be welcome. Is there a simple way to explicitly parameterize bricklayer circuits such that some symmetries are realized? And could a simple numerical test of such a parameterization be performed?

  6. Figures 9-11 are all referenced in the main text, but are relegated to appendices. For casual readers it makes sense to shift these figures to the main text to emphasize their relevance to the author's discussion.

Requested changes

  1. Clarify discussion of gradient descent; provide enough detail for other researchers to go out and verify results.

  2. Clear up the discussion of figure 5, ideally by giving explicit gate counts.

  3. For casual readers, the use of the word "exact" will be confusing. Consider reading points in which you use that word as a short-hand, and being more explicit where it makes sense so that the reader understands detailed comparisons made.

  4. Additional discussions of the role of lattice geometry seem worthwhile, and perhaps even essential in other quantum simulation contexts.

  5. Additional discussion of the use of symmetries in the context of the author's compiled circuit ansatz.

  6. Move figures 9-11 to the main text.

  • validity: high
  • significance: good
  • originality: good
  • clarity: good
  • formatting: excellent
  • grammar: excellent

Author:  Maurits Tepaske  on 2022-10-24  [id 2950]

(in reply to Report 2 by Michael Flynn on 2022-10-02)

We thank the referee for his thorough report of our study. Below we react to his remarks.

  1. We agree with this remark on the lack of optimization details. We have added an appendix where we discuss the optimization procedure in more detail, as well as the challenges that are faced during the optimization.
  2. We agree and now explicitly mention the corresponding gate counts to the discussion of these figures.
  3. We agree with this statement, the "exact" unitaries we are approximating are for L>8 in fact finite bond-dimension approximations of the exact time evolution operators. To emphasize this we now mention this fact early in the manuscript and more appropriately now name it the "targeted" time evolution operator or unitary in the remainder.
  4. We thank the referee for bringing up this point. However, we are unable to track this difference in scaling of the stacking, between the chain and ladder, to a difference in the model geometry. We have checked that for timestep dt the infidelity at time n_stacks*dt<<1 scales as dt^2, for both the chain and ladder geometries, as expected for the second-order Trotter decomposition, but for the stacking at fixed dt and intermediate stacked time (n_stacks*dt of order 1) we obtain the different scalings observed in Fig. 6. It could be that the sudden change at n_stacks*dt ~ 1 is due to a reordering of the error terms in the stacked Trotter decomposition, or that we are simply observing a crossover between proper Trotter scaling and the error saturation regime. We have attached a plot to this reply where this can be seen. Here the same setup as in Fig. 6 is used, but now we show also some values outside of the range shown in the paper (which is demarked with dashed vertical lines).
  5. In the new version we have added an appendix with a discussion on lattice symmetries for our circuits. For the models considered in our study this boils down to lattice inversion symmetry. We now show a plot which characterizes whether our general circuit ansatz and the used optimization procedure yield circuits that respect this symmetry on the gate level. Also, we mention how this symmetry can be incorporated in the circuit ansatz.
  6. While we indeed mention these figures in the main text, we think that they are not of vital importance to the main storyline and instead would result in clutter. However, if the referee strongly disagrees with this we are willing to put them in the main text.

Attachment:

exact_trotter_eps_L12.pdf

---

## Round 2 · Referee Report · Subhayan Sahu (Referee 1) · 2022-11-22

Report

The authors have sufficiently modified the paper to address the points raised in the previous report.

---

## Round 2 · Referee Report · Michael Flynn (Referee 2) · 2022-11-29

Report

I am satisfied by the author's responses to previous requests for edits, and in their responses to more detailed questions. I'm happy to recommend this for publication.

---

## Round 2 · List of Changes

• Added appendix with details on optimization.
  • Added appendix on the learning of Hamiltonian lattice inversion symmetry.
  • Replace absolute OTOC values in main text with relative errors, which is now moved to the appendix with the other OTOC plots.
  • Added plot of OTOC refocussing to appendix.
  • Small changes to the text, e.g. gate counts of used circuits now explicitly mentioned.

---

## Editorial Decision

published